# The Integration of Gender Perspective into Young People’s Sexuality Education in Spain and Portugal: Legislation and Educational Models

**DOI:** 10.3390/ijerph182211921

**Published:** 2021-11-13

**Authors:** Aliete Cunha-Oliveira, Ana Paula Camarneiro, Sagrario Gómez-Cantarino, Carmen Cipriano-Crespo, Paulo Joaquim Pina Queirós, Daniela Cardoso, Diana Gabriela Santos, María Idoia Ugarte-Gurrutxaga

**Affiliations:** 1Health Sciences Research Unit: Nursing (UICISA: E), Coimbra School of Nursing (ESEnfC), 3004-011 Coimbra, Portugal; pcamarneiro@esenfc.pt (A.P.C.); sagrario.gomez@uclm.es (S.G.-C.); pauloqueiros@esenfc.pt (P.J.P.Q.); dcardoso@esenfc.pt (D.C.); dianagabrielasantos@gmail.com (D.G.S.); 2Research Group Nursing, Pain and Care (ENDOCU), University of Castilla-La Mancha (UCLM), Av. Carlos III s/n, 45071 Toledo, Spain; maria.ugarte@uclm.es; 3Faculty of Physiotherapy and Nursing of the Toledo Campus, University of Castilla-La Mancha (UCLM), 45071 Toledo, Spain; 4Faculty of Health Sciences, University of Castilla La Mancha, 45600 Talavera de la Reina, Spain; mariacarmen.cipriano@uclm.es; 5University and Hospital Center of Coimbra, 3004-561 Coimbra, Portugal

**Keywords:** sex education, adolescent, gender analysis, education, legislation, primary health care

## Abstract

Throughout history, Sexuality Education (SE) has undergone many changes in formal education curricula. The education systems should incorporate SE and promote an understanding of sexuality from the critical perspective of gender. Objectives: To examine the approach to SE in young people in Spain and Portugal, considering the incorporation of the gender perspective, and analyze the legislation in both countries. A scoping review was conducted considering studies with SE programs, gender perspective, and legislation in Spanish, Portuguese, and English, without any time limits. The population consisted of young people aged 10 to 18 years who did not attend higher education. Databases used: CINAHL Complete, ERIC, LILACS, SciELO, MEDLINE, Psychology and Behavioral Sciences Collection, Scopus, Open Access Scientific Repository of Portugal, *Base de Datos de Tesis Doctorales*, Theses and Dissertations Online, and governmental websites. Thirty-two studies were found, including intervention, diagnosis, and documental programs. Eight of the studies adopted the gender perspective. Legislation in both countries is vast, with 23 main references. Although SE is legislated in both countries, the social-health and educational programs are insufficient. The relevance of the gender perspective is not incorporated into SE.

## 1. Introduction

The right to education in Spain is recognized by the Spanish Constitution of 1978 and in Portugal by the Portuguese Constitution of 1976. The objective of education is the full development of personality, including the dimension of sexuality, which justifies the integration of sexuality education into the educational process; however, although different legal milestones have resulted in different ways of developing sexuality programs, the education system’s approach to sexuality education continues to be a pending issue.

The implementation of sexuality education in Spain and Portugal has been a long and complex process, influenced by the similarities between the political regimes of both countries. Both went through a period of dictatorship and repression during which people had no freedoms or rights, let alone the right to sexuality education. Moreover, these political regimes were closely linked to the Catholic Church. Schools were confessional, many of them still are today, and the Catholic Church was very conservative in matters of sexuality. Both ideologies left their mark on the educational models, which lacked content related to sexuality education. When there was some content on this subject, it only addressed physiological functions. Thus, the implementation of sexuality education in classrooms was slow and practically non-existent [1].

According to the World Health Organization (WHO), affective-sexuality education is a right of children and adolescents, together with other sexual rights such as freedom, privacy, equity, coexistence based on equality, and without any form of discrimination. Sexuality is a core dimension because it is present throughout our lives. It is an interaction of biological, psychological, social, political, economic, cultural, legal, ethical, historical, religious, and spiritual dimensions [2].

One of the areas where sexuality is most appropriately addressed is education. According to Pellejero Goñi and Torres Iglesias [3], sexuality education must be framed within the perspective of sexology and based on coeducational approaches that treat people holistically. Thus, sexuality education, or affective-sexual education, should be integrated as knowledge into all basic educational stages because education is founded on the principle of respect for human rights and promotes relational ethics, which is a key factor for a society to strengthen its democratic principles of peace, freedom, equality, tolerance, and solidarity [4].

According to UNESCO [5] (p. 12), “The lack of Sexuality Education in schools, as well as inadequate Sexuality Education can harm the development and well-being of children and adolescents, particularly in those contexts where misinformation-misuse of information, beliefs, prejudices, cultural and social norms and practices-create a situation of vulnerability to issues such as pregnancy, motherhood and fatherhood in adolescence, sexual violence, the human immunodeficiency virus (HIV), and other sexually transmitted infections (STIs).”

Although the concept of sexuality has gradually come to include not only biological but also social, emotional, and psychological aspects, among others, the biological-hygienist approach is still present in educators’ practices in traditional schools [6]. This is a limited concept that rarely covers other issues observed in social relations such as gender violence, gender equity, gender stereotypes, sexual violence, among others [6]. Scientific evidence suggests that Comprehensive Sexuality Education (CSE) programs are the most useful compared to programs based on moral/traditional models or focused on “abstinence” [7]. International studies have shown that CSE is associated with increased knowledge, better Sexual and Reproductive Health (SRH) outcomes, and less risky practices. Furthermore, CSE does not increase sexual activity, delays the age of sexual initiation, reduces the number of sexual partners, and improves STI protection practices, being an effective strategy to reduce HIV-related risks, increase condom use [8], and prevent unintended pregnancies [9].

In 2018, UNESCO published a technical report entitled *International Technical Guidance on Sexuality Education*: An evidence-informed approach that argues for the importance of delivering CSE to young people to achieve the Sustainable Development Goals. According to this document, “Comprehensive sexuality education (CSE) is a curriculum-based process of teaching and learning about the cognitive, emotional, physical, and social aspects of sexuality. It aims to equip children and young people with knowledge, skills, attitudes, and values that will empower them to: realize their health, well-being, and dignity; develop respectful social and sexual relationships; consider how their choices affect their well-being and that of others; and understand and ensure the protection of their rights throughout their lives” [10] (p. 16). Therefore, CSE goes beyond education about reproduction, risks, and disease and must respect the sexual rights of boys and girls. Based on this document, CSE is delivered in formal and non-formal settings based on a written, comprehensive curriculum that focuses on human rights and gender equality. CSE addresses the different ways gender norms can influence inequality and how inequality can influence children and young people’s overall health and well-being while also impacting efforts to prevent HIV, STIs, unintended pregnancies, and gender-based violence.

A gender-sensitive approach to CSE aims to foster respect for others and a relationship between men and women based on co-responsibility [11,12]. This procedure is very relevant considering the lack of information among young people in Europe [13], the maintenance of the sexual double standard [14], and the inequalities in the assumption of responsibilities for the prevention of STIs and unintended pregnancies among girls and boys [15].

Gender is a social construct that has its meaning but is also influenced by cultural factors. For Singer Kaplan [16], it is one of the most important factors influencing people’s attitudes towards sexuality and cultural influences. Gender inequality increases both female and male risk-taking behaviors. According to the Joint United Nations Programme on HIV/AIDS [17], in unprotected heterosexual intercourse, women are twice as likely as men to acquire HIV from an infected partner. Adolescent girls and women are particularly vulnerable to HIV, the cause of gender inequality that threatens their lives. Thus, the CSE model should go beyond prevention. It should establish a biopsychosocial perspective of sexuality based on biological factors and their interaction with psychological and socio-cultural factors, including people’s biological, psychological, and socio-affective dimensions [18].

Gender as a category of analysis allows identifying the psychosocial determinants of sexual attitudes and behaviors that cause harm to health [19,20]; however, this model of sexuality education has not always been like that.

The fundamental question that arises today is how we can know the evolution of the gender perspective considering the approach to sexuality education in schools and adolescent health care in Spain and Portugal. WHO limits adolescence to the second decade of life (10–19 years old) and defines youth as the 15–24-year age group. These concepts have been developed to identify older adolescents (15–19 years old) and young adults (20–24 years old) [21]. A search conducted before the review on several databases did not find any review that could answer the question set out.

This review aims to explore the approach to sexuality education in the education of young people in Spain and Portugal, considering the incorporation of a gender perspective, and analyze the legislation in both countries.

## 2. Materials and Methods

### 2.1. Study Design

A scoping review was conducted based on the Joanna Briggs Institute (JBI) methodology and the Preferred Reporting Items for Systematic Reviews and Meta-Analyses: Extension for Scoping Reviews (PRISMA-ScR) Checklist [22].

### 2.2. Inclusion Criteria

The inclusion criteria were established according to the PCC mnemonic for scoping reviews (Population, Concept, Context). The population of interest (P) consists of adolescents aged 10–19 years attending non-higher education. The concept (C) considers the legislation and curricular and health programs on sexuality education from the gender perspective. The context (C) considers the educational and primary health care settings in Portugal and Spain.

All study designs were considered. Studies written in Portuguese, English, and Spanish were considered. No time limits were imposed on the studies.

### 2.3. Search Strategy

Initially, a search was conducted in two databases (MEDLINE via PubMed and CINAHL via EBSCO) to analyze the natural language and descriptors used in the identified studies. Subsequently, between January and April 2021, a complex search was conducted in published literature databases, namely CINAHL Complete via EBSCO, ERIC via EBSCO, LILACS, SciELO, MEDLINE via PubMed, Psychology and Behavioral Sciences Collection via EBSCO, and Scopus. Grey literature was searched via the Open Access Scientific Repository of Portugal (RCAAP), *Base de Datos de Tesis Doctorales*, and Theses and Dissertations Online.

A search was performed on governmental websites to identify the legislation in Portugal and Spain. Finally, resources obtained through a manual search were also used. The search strategies for each resource are listed in Appendix A (Table A1, Table A2, Table A3, Table A4, Table A5, Table A6, Table A7, Table A8, Table A9 and Table A10).

### 2.4. Data Analysis

The selection process was carried out by two independent reviewers who screened the titles and abstracts and read the full articles, theses, and legislation that met the inclusion criteria. Two independent reviewers extracted the data. Any disagreements were resolved with a third reviewer. There was no need to contact the authors of the primary studies for clarification.

The data extraction process took into account: (1) author and year of publication; (2) country where the study was conducted and language; (3) type of study design; (4) population characteristics; (5) context; and (6) incorporation of the gender perspective.

## 3. Results

The search for published and unpublished studies found 1237 records (n = 386 in CINAHL Complete, n = 3 in ERIC, n = 133 in LILACS, n = 130 in SciELO, n = 191 in MEDLINE, n = 29 in Psychology and Behavioral Sciences Collection, n = 168 in Scopus, n = 75 in RCAAP, n = 120 in *Base de Datos de Tesis Doctorales*, and n = 2 in Theses and Dissertations Online).

After duplicates were removed (n = 241), 996 records were identified. A total of 947 records were excluded after title and abstract screening. Then, the full text of all included records was obtained (n = 49). The full text of these 49 studies was reviewed: 26 studies were included, and 23 were excluded. The reasons for exclusion were as follows: population of interest (n = 2); concept under study (n = 10); context of interest (n = 7); concept and context (n = 2); population and context (n = 1); abstract only (n = 1).

Other records were identified through other search methods: six were obtained from a manual search, and 23 legislative documents were obtained from governmental websites (Portuguese websites n = 11; Spanish websites n = 12).

Thus, 26 studies obtained through the systematic search and six manually obtained studies (n = 32) were screened, plus 23 legislative records, in a total of 55 studies/documents included in this review. Figure 1 shows the flow diagram of the study selection and inclusion process [23].

### 3.1. A general Overview

The studies on sexuality education models in Spain and Portugal were published between 1990 and 2020. They are presented in chronological order of publication and summarized in Table 1 and Table 2, respectively.

Twenty-one studies were published in Spain (Table 1), distributed as follows: four articles were published in the 1990s, with the first study being published in 1990, five studies were published between 2000 and 2010, and 12 studies were published between 2011 and 2020. Eleven studies were published in Portugal (Table 2), distributed as follows: the first study was published in 2007, followed by ten studies published in 2011–2020, with the last study published in 2019.

The analysis of the context of the included studies and their characterization shows that the Spanish studies were conducted in the following contexts: School health/Educational context, 2; educational context, 17; social-health care context, 1; primary health care, 1. In Portugal, four studies were conducted in the School health/Educational context, and seven were conducted in the educational context.

Concerning study design, 10 were qualitative studies, 17 were quantitative studies, 1 was a mixed-methods study, and 4 were theoretical studies. Eight articles referred to gender differences (5 in Spain and 3 in Portugal).

In the Spanish context, this review found theoretical studies based on documentary analysis [24,25,26,27]. Fernández Costa, Juárez Martínez, and Díez David [25] concluded that specific HIV/AIDS prevention programs predominate over affective-sexual programs in almost all autonomous communities in Spain. In a study on the analysis of sexual health content aimed at young people on the websites of official institutions, Conesa et al. [27] showed that the sexual health model being promoted is based on risk prevention, namely on reducing the risk of acquiring STIs or unintended pregnancies. De Irala, Gómara Urdain, and Del Burgo [24] analyzed the content of 12 textbooks published in 2002 on sexuality and human reproduction and found inaccurate information. On average, 12.6 incorrect messages were identified in each textbook. Rascón and Sandoica [26] analyzed epidemiological information on the impact of adolescent sexual practices on the health sector (abortion, unintended pregnancies, and STIs), questioning the validity of “safe sex” campaigns from the perspectives of ethics and health care efficiency.

Gómez-Galán et al. [28] and Guerra Marmolejo [29] analyzed sexuality education among young people in the educational setting. Gómez-Galán et al. [28] highlighted young people’s lack of knowledge about sexuality, reproduction, and risky behaviors in sexual intercourse. They emphasized the need for faculty training in this area. In the province of Malaga, Guerra Marmolejo [29] concluded that the sexuality education delivered to adolescents in Malaga was not entirely adequate, holistic, and healthy.

Eight studies analyzed the effectiveness of sexuality education programs [30,31,32,33,34,35,36,37]. Sevilla Heras [37] presents the “SOMOS”—sexuality education program, aimed at 3rd-year students of Compulsory Secondary Education (ESO) in Spain. Its main objective is to develop positive attitudes towards sexuality associated with healthy and satisfactory sexual experiences. Escribano [31] and Morales et al. [35] have assessed the results of the implementation of COMPAS (Competencies for Adolescents with a Healthy Sexuality), a program focused on education and training in condom use among young people. In both studies, the findings support the importance of implementing sexuality education interventions before adolescents have their first intimate relationships to achieve a greater impact. Further, also related to condom use as the best strategy to prevent HIV/AIDS transmission, Morales et al. [34] assessed the results of the implementation of the ¡Cuídate! program and concluded that it positively affected consistent condom use.

At the regional level, more specifically in Asturias, García-Vázquez et al. [32] analyzed the results of the development of a secondary school program called *Ni Ogros ni Princesas* (Neither Ogres nor Princesses) that educates students for four years with trained teachers and external workshops. In his thesis, Sanchez Ramiro [36] analyzed the effects of a sexuality education program called ‘Face to Face, Heart to Heart’ and concluded that it effectively reduces risky sexual behaviors among young people.

Three studies showed the results obtained over three years by comparing data from a follow-up assessment of the effects of COMPAS on adolescent sexual risks and behaviors with an evidence-based intervention (¡Cuídate!) and a control group [38,39,40].

Concerning the incorporation of the gender perspective, five studies addressed this issue [27,32,36,37,41]. These studies highlight the need to incorporate the gender perspective in sexuality education for reducing the cognitive, attitudinal, and emotional variables of gender inequalities in boys and girls.

**Table 1 ijerph-18-11921-t001:** Characteristics of the studies conducted in Spain.

Authors/Year	Objectives	Design	Context	Gender Perspective
Calles et al., 1990 [42]	To increase young people’s knowledge/raise awareness about sexual health, birth control methods, and sexual health consultations in primary health care.	Qualitative	Educational/School Health	No
Amores et al., 1998 [43]	To analyze the ideal characteristics of sexuality consultations for young people.	Theoretical	Primary Care	No
Fernández Costa et al., 1999 [25]	To describe and assess the affective-sexual programs published in Spain (1990–1997).	Theoretical	Educational	No
Vaqué et al., 1999 [44]	To promote a positive and responsible attitude towards sexual health.	Qualitative	Educational/School Health	No
Gómez-Galán et al., 2003 [28]	To evaluate the objectives, contents, and methodology used in the sex education programs in Merida; To evaluate students’ knowledge, attitudes and satisfaction.	Qualitative (focal and nominal groups)	Educational	No
De Irala et al., 2008 [24]	To analyze the content of textbooks in the areas of sexuality and reproduction; To evaluate the extent to which these textbooks promote healthy reproductive lifestyles.	Qualitative	Educational	No
Rascón et al., 2008 [26]	To analyze sexual education campaigns in Spain from an ethical perspective.	Theoretical	Social-Health care	No
Climent et al., 2009 [41]	To describe the different prevailing approaches to the sexual education of adolescents who became pregnant in the context of a particular gender socialization and its relationship with some reproductive behaviors.	Qualitative	Educational	Yes
Hernández-Martinez et al., 2009 [33]	To evaluate the effectiveness of a sex education program in the acquisition of knowledge about contraceptive methods and emergency contraception, as well as in changing attitudes to condom use and responsible use of emergency contraception.	Quantitative	Educational	No
Claramunt Busó, 2011 [30]	To design a sexual education program (PESex) and assess its effectiveness in adolescents.	Quantitative	Educational	No
Sevilla Heras, 2011 [37]	To assess the effectiveness of the SOMOS program: attitudes towards sexuality, attitudes towards masturbation, knowledge and beliefs about sexuality, adherence to the sexual double standard; satisfaction with the program.	Quantitative (quasi-experimental)	Educational	Yes
Escribano et al., 2015 [31]	To determine the factors that mediate the consistent use of condoms over the 24-months post-intervention period in adolescents who received the COMPAS program (*Competências para adolescentes con una sexualidad saludable*).	Quantitative (RCT pre-post 12–24 months)	Educational	No
Espada et al., 2015 [38]	To evaluate the efficacy of the COMPAS program compared with a Spanish-culture adapted version of ¡Cuídate!, an evidence-based HIV-prevention curriculum.	Quantitative (RCT)	Educational	No
Morales et al., 2015 [40]	To compare data from a 12-month follow-up of the effects of COMPAS, ¡Cuídate!, and control group on sexual risks and sexual behaviors.	Quantitative (longitudinal)	Educational	No
Espada et al., 2016 [39]	To assess the effects of COMPAS and compare it with an evidence-based program (¡Cuídate!) and control group.	Quantitative (3-arm RCT)	Educational	No
Guerra Marmolejo, 2017 [29]	To explore whether the sexuality education (concepts, attitudes, risky behaviors, STIs, gender-based violence) delivered to adolescents in Malaga is adequate, holistic, positive, and healthy.	Quantitative (ex post facto)	Educational	No
Morales et al., 2017 [34]	To identify mediators of the intervention’s effects compared to a control group.	Quantitative (RCT)	Educational	No
Conesa Rodríguez et al., 2018 [27]	To analyze the visual and written content of the Sexual Health Portal for young people and compare it with the websites on sexual health from official institutions.	Qualitative	Educational	Yes
Sanchez Ramiro, 2018 [36]	To determine the effects of the sexuality education program ‘Face to Face, Heart to Heart’ and analyze the evolution of knowledge and attitudes towards sexuality considering the gender perspective.	Quantitative (Quasi-experimental)	Educational	Yes
García-Vázquez et al., 2019 [32]	To apply the ‘Neither ogres nor princesses’ program in Asturias.	Quantitative (Quasi-experimental, longitudinal)	Educational	Yes
Morales et al., 2020 [35]	To analyze psychosocial and behavioral changes (sexual experience) at baseline, post-test, and 12- and 24-month follow-ups.	Quantitative (Cluster RCT)	Educational	No

In the Portuguese context, there are three documentary analysis studies [6,45,46]. Gama and Anastácio [6] pointed out that, in Portugal, sexuality education is mandatory after the 1st cycle of basic education since the approval of Law No. 60/2009 of 6 August. Moreover, Vilaça [46] highlights the socio-pedagogical milestones of the implementation of sexuality education in Portugal since 1970. The compulsory nature of sexuality education in Portugal enabled a systematic and specific work in addressing sexuality and gender issues, monitoring and evaluating the work in schools. Siqueira and Netto [45] concluded that there are advances and limitations in the current reality of sexuality education in school settings. The Portuguese government is responsible for providing specific training to teachers, who work under the responsibility of the Ministry of Education rather than the Ministry of Health and advancing against the pre-established structures of hygienists, reflecting on, assessing, and monitoring social changes.

Four studies addressed the application of sexuality education programs [47,48,49,50]. These studies assessed the effects of sexuality education programs, namely increased knowledge about family planning [47], STIs [47,48,50] and contraceptive methods [48,50], and more liberal attitudes towards sexuality [49].

Concerning the studies that diagnosed the situation [51,52,53,54]. Silva and Carvalho [52] reported that sexuality education curricula focus on the biopsychosocial or personal-social development models but do not establish an embodied approach. Rocha et al. [53] found that 96% of the Portuguese schools included in the study sample implemented some form of sexuality education. According to Matos et al. [51], the school directors rated the implementation of Law no. 60/2009 of 6 August on the contents of sexuality education as good/very good; however, in a sample of 43 nurses, Vilaça [54] mentions that teachers still lack knowledge about reproductive health.

Concerning the approach to gender issues, three studies addressed this phenomenon of interest [6,50,52]. Gama and Anastácio [6] addressed the need for systematic, specific work on sexuality and gender issues in schools in Portugal. Silva and Carvalho [52] mentioned the existence of resources available for didactic support to interventions on gender and sexual orientation. Escalhão [50] highlighted the importance of demystifying gender roles (see Table 2.)

**Table 2 ijerph-18-11921-t002:** Characteristics of the studies conducted in Portugal.

Authors/Year	Objectives	Design	Context	Gender Perspective
Sousa et al., 2007 [48]	To assess sexual knowledge, attitudes, and experiences in two groups of youth: 1- participated in the Experimental Project of Sex Education and Health Promotion in Schools (EPSEHPS); 2- did not receive any formal sex education.	Quantitative	Educational/School Health	No
Ribeiro et al., 2012 [49]	To evaluate the impact of a sexual education program in the classroom on adolescents’ attitudes towards sexuality.	Quantitative (quasi-experimental)	Educational/School Health	No
Vilaça, 2013 [46]	To describe the evolution of the public policies and educational practices of sexuality education in the school community.	Theoretical	Educational	No
Matos et al., 2014 [51]	To assess the implementation of sexuality education in school settings.	Quantitative	Educational/School Health	No
Silva & Carvalho, 2014 [52]	To analyze the relevance of implementing an embodied sexuality education and how bodies are addressed in the current sexuality education curriculum. To analyze the evolution of the sexuality education curriculum in terms of paradigms, discourses and practices.	Qualitative (Case study)	Educational	Yes
Escalhão, 2015 [50]	To design, implement, and assess training strategies for students in the 7th to 12th grades, enabling informed and safe choices and the adoption of health behaviors. To demystify concepts and/situations related to contraception, STIs, affectivity, and gender roles.	Mixed (Qualitative and Quantitative)	Educational	Yes
Rocha & Duarte, 2015 [53]	To examine in Portuguese schools the facilitating factors at micro- and exosystem levels associated with more effective implementation of sexuality education.	Quantitative	Educational/School Health	No
Almeida, 2016 [47]	To determine the effectiveness of an educational intervention in the context of sexuality in the knowledge about STIs, family planning, and attitudes towards sexuality.	Quantitative	Educational	No
Siqueira & Netto, 2018 [45]	To investigate official documents that regulate sexuality education in Brazil and Portugal.	Qualitative (documental)	Educational	No
Gama & Anastácio, 2019 [6]	To reflect on the regulation of sexuality education and the teacher’s legitimacy in Basic Education to address issues related to sexuality and gender.	Qualitative (documental)	Educational	Yes
Vilaça, 2019 [54]	To understand the teachers’ practices to develop guidelines for their in-service training on sexuality education.	Qualitative	Educational	No

### 3.2. Legislation in Spain and Portugal

Below is the legal framework for sexuality education in Spain (Table 3) and Portugal (Table 4) in educational and non-educational areas. The first legislative instruments on sexuality education date back to 1984 and 1985, respectively, in Portugal and Spain.

In Spain, the right to education was regulated in 1985, with the Organic Law 8/1985 (LODE) [55]. After this law, sexuality education was regulated through the publication of laws, royal decrees, and regional equal opportunities strategic plans aimed to protect women and fight against male chauvinist domestic violence.

Organic Law 1/1990 of 3 October (LOGSE) [56], on the general organization of the education system, included sexuality education in the different stages of the education system. Organic Law 10/2002 of 23 December on the Quality of Education (LOCE) [57] eliminated sexuality education and, as a step backward, it was never implemented. The focus on gender equality and coeducation in all stages of learning appears for the first time in Organic Law 2/2006 on Education (LOE), of 3 May [58], which revoked the previous law, constituting an important advance in the regulation of this matter. It also includes equality between men and women, the prevention of gender violence, and the respect for affective-sexual diversity, introducing educational and professional guidance for students with an inclusive and non-sexist perspective in secondary education. It incorporates the “recognition of affective-sexual diversity” and maintains sexuality education as a transversal subject (as in LOGSE). It incorporates the term sexuality education, addressing it in an interdisciplinary way under the influence of the Law against Gender Violence. With this law, the subject “Education for Citizenship” was included, which addressed sexuality education but was subjected to the schools’ will. This subject was surrounded by controversy on the part of the Family Forum and the National Catholic Confederation of Parents and Students. Organic Law 3/2007 of 22 March [59] on the equality between women and men refers to the different dimensions of Sexual and Reproductive Health (SRH): motherhood, fatherhood, leaves, and sexual harassment.

The 2008–2011 Equal Opportunities Strategic Plan aims to improve women’s health and impact gender-sensitive diseases, namely to include the prevention of heterosexual transmission of HIV/AIDS. It also aims for sexual and reproductive health programs in the services provided by the National Health System. These programs develop interventions to provide information and sexuality education, prevent unintended pregnancies, and promote access to adequate and effective birth control methods [60].

Sexuality education is a subject recommended by the Spanish State in Organic Law 2/2010 of 3 March on sexual and reproductive health and the voluntary interruption of pregnancy [61]. It refers to affective-sexual and reproductive education in the formal context of the education system, but it is not compulsory. Organic Law 8/2013 of 9 December on the improvement of the quality of education (LOMCE) was a setback because it did not include sexuality education in the curricula [62]. Sexuality education was not included either as a compulsory subject or as an optional subject, nor was it explicitly included in the existing areas of knowledge or basic competencies to be acquired by students. Thus, sexuality education was only taught in the formal context in some cases by certain teachers during the hours assigned to tutoring, with the responsibility being delegated to each school and the approach and the content being chosen by the teacher. Overall, the contents are addressed from a heterosexual approach that limits sexuality to the reproductive stage and directs the information exclusively towards risk prevention, particularly unintended pregnancies, STIs, and HIV.

Due to these legislative shortcomings at the state level, some autonomous communities, using their competencies in education, incorporated this topic in their schools, such is the case of the SKOLAE Program, the Coeducation Plan 2017–2021 for the schools and educational communities of Navarre, that incorporates “learning about sexuality and proper treatment (...), self-knowledge towards the construction of relationships and love based on acceptance and respect for diversity, free from chauvinist violence” [63].

The most recent law is Law 4/2018 of 8 October, for a Society Free of Gender Violence in Castile-La Mancha. According to Title II. Prevention and Awareness; Chapter I. Education; Article 9. Educational interventions: “1. The Regional Government of Castile-La Mancha shall create, within its competencies, a compulsory subject with contents related to equality, affective-sexual education, and prevention of gender-based violence to be taught both in Primary Education and in Compulsory Secondary Education in order to transmit the values of equality, respect, and diversity. It introduces affective-sexual education on a cross-cutting basis in the syllabus of all subjects, elements that value equality, promoting women’s visibility, and, in history, the origin, development, and achievements of the feminist movement and women’s history” [64].

In the II Equal Opportunities Strategic Plan for Men and Women in Castile-La Mancha (2019–2024), one of the priority areas for action was affective-sexual education and the prevention of violence against women [65].

Organic Law 3/2020 of 29 December (LOMLOE), amending Organic Law 2/2006 on Education, of 3 May, entered into force in January 2021. This law adopted a gender equality approach through coeducation, including, among its principles and aims, the adaption of affective-sexual education to students’ level of maturity and the prevention of gender violence. It also includes affective-sexual education in the compulsory subject of Health Education and as a cross-cutting subject, both for primary and secondary education [66].

Since the 1980s, Portugal has published laws, decree-laws, ordinances, and orders that constitute important milestones in regulating sexuality education (Table 4).

The 1st specific Portuguese law on sexuality education and family planning in sexuality education in the school community was Law 3/84 [67]. This law created, through the Ministry of Health, the free youth care centers and foresees that the government develops measures concerning family rights, family planning, and the adequacy of the training curricula of health professionals in the area of sexuality. In this law, sexuality education became a right secured by the government. It established that the Portuguese government was responsible for “securing the right to sexuality education as a component of the fundamental right to education” [46]. This law also established that basic and secondary education curricula should include scientific knowledge about anatomy, physiology, genetics, and human sexuality. Ordinance 8/1985 [68] of the Ministry of Health approved the regulation of family planning consultations and youth care centers. This legislation is the starting point for the interaction between health and education services for young people’s sexuality education.

The Basic Law of the Portuguese Educational System (Law 46/86) [69] included sexuality education in a new, cross-cutting subject entitled Personal and Social Development, to which components such as family education and health education were associated. Later, with Decree-Law no. 286/89 of 29 September [70], the School-Area was created to materialize the subject of Personal and Social Development in the curricula, with 95–110 h per year, not disciplinary, under each school’s responsibility.

In the 1990s, Order 172/ME/93, of 13 August [71], created the Program for Health Promotion and Education, which became a support and resource tool for schools, mainly in response to problems and needs in the areas of drug addiction and AIDS prevention, sexuality, and development of personal and social skills, among others. It ended in 1999 after Portugal joined the European Network of Health Promoting Schools [46]. Given this evolution and to meet the needs for information on this topic, especially among young people and adolescents, Law no. 120/99 [72] called for the joint work between the Ministry of Education and the Ministry of Health and created measures to promote sexuality education, reproductive health, and the prevention of STIs, as well as to facilitate the voluntary interruption of pregnancy in cases where it is legally permissible. In this way, it promoted a compulsory, comprehensive, and cross-cutting approach to sexuality education, involving students, tutors, and their associations, and teacher training. This law created the necessary dynamics for the inclusion of sexual and reproductive education in the curricula. It was the first law to clearly introduce the issue of gender equality.

The regulation of Law 120/1999 [72] by Decree-Law 259/2000 of 17 October [73] established that the school is the competent institution to integrate sexual health promotion strategies in the development of the curricula and the organization of curricular enrichment activities, promoting the school–family relationship. Thus, it included sexuality education and gender equality in primary and secondary education curricula, integrated into health education. The obligation for schools to include the area of health education in their educational project, combining disciplinary transversality with thematic inclusion in the non-disciplinary curricular area and creation of a territorially balanced and efficient network of educational, social, and psychological resources to support schools and teachers was the object of Order no. 25 995/2005 (2nd series), of 16 December [74]. However, in 2006, a new Order was published (Order 15987/2006) [75], reaffirming that sexuality education in schools would become one of the four components of the Health Education Project that all schools should prepare and implement. This project would be coordinated by a teacher appointed by the school for Health Education, in accordance with Order no. 2506/2007 [76].

Law no. 60/2009 [77] established the regime for the implementation of sexuality education in school settings. Presentation of the contents of sexuality education foreseen in the law and the recommended workload. Sexuality education became mandatory in the educational projects of school clusters and non-clustered schools, with a workload adapted and distributed by each educational level, specified for each class, and distributed in a balanced way throughout the several periods of the school year—as a model of psychosocial sexuality education or biopsychosocial sexuality education. Ordinance no. 196-A/2010 [78] regulated and established the regime for the implementation of sexuality education in school settings, including the contents to be taught in each study cycle.

## 4. Discussion

### 4.1. Legislation

The right to education was approved in Spain for the first time in 1985. However, it was not until the enactment, five years later, of the law on the general organization of the education system (Organic Law 1/1990) [56] that sexuality education was included for the first time in the different educational stages and through the different areas in a cross-cutting way. Although there is no mention of gender, it reflects an important social moment.

In Portugal, the Constitution of the Portuguese Republic created, in April 1976, the legal basis for the promotion and education for sexual health in the country, with schools becoming the educational space for promoting students’ participation and their academic, personal, and social development [46]; however, it was only in the 1980s that the debate started about sexuality and sexuality education in schools as citizens’ rights. In early 1984, the first legal document about sexuality education and family planning in schools [67] advocated the inclusion of scientific knowledge about human anatomy, physiology, genetics, and sexuality, adapted to the different levels of education; however, the component related to sexuality education in schools was not regulated, and the components related to health services and family planning, namely family planning consultations and youth care centers, were regulated in the following year [68].

At the same time, the ongoing processes of reorganization of basic education and the revision of the curricula in secondary education gave special attention to the need for an integrated approach to this subject as an essential dimension of the education and training of Portuguese youth. The Basic Law on the Educational System (Law no. 46/86) [69] included sexuality education in a new, cross-cutting area (Personal and Social Development). It established that the school had the right and duty to promote sexuality education. Article 28 (Support to School Health) stated that: “students’ healthy growth and development shall be monitored by specialized services of primary health care”. The inter-ministerial report for the elaboration of the Action Plan on sexuality education and family planning, approved in October 1998, put forward some concrete measures for applying the 1984 law and understood sexuality education as “an essential component of health education and promotion.”

Youth Care Centers were created to respond to situations related to adolescents’ personal development, particularly concerning sexual and reproductive health, family planning, STIs, and prevention of psychoactive substance use and other risk behaviors. These centers are supported by interdisciplinary teams providing prevention and information interventions, counseling, medical, nursing, psychological support, and personalized counseling in specialized consultations. These consultations are free of charge, following Article 6 of Law no. 3/84 [67], on sexuality education and family planning, which allows it to be an open door for the 11–25-age group, with a special focus on sexual and reproductive health, offering all consultations and condoms free of charge, regardless of the use of other birth control methods.

Although the regulation of Law 120/99 [72] created the necessary dynamics for the inclusion of sexual and reproductive education interventions in the curricula, in the 2000/2001 school year, most Portuguese schools still did not include sexuality education in their educational project [46]. It was the health-promoting schools that had a major positive influence on the future of sexuality education in Portugal up to the present day [46]. It should be noted that Portugal remains one of the countries with a high incidence of HIV infections among the 25–29 age group in the European Union [79].

In Spain, at the turn of the millennium, some education reforms had a negative impact on sexuality education due to political changes. The Organic Law on the Quality of Education [57] eliminated sexuality education. It was a step backward, as it promoted an exclusively reproductive vision of sexuality; however, this law was never implemented. As a result of another political change, the Organic Law on Education 2/2006 [58] incorporated the “recognition of affective-sexual diversity” and maintained sexuality education as a cross-cutting subject, as in LOGSE [56]. It incorporated the subject Education for Citizenship, so the approach to sexuality education was always subjected to the schools’ will.

Since the publication of this law, other laws, royal decrees, and regional equal opportunities strategic plans aimed at protecting women and fighting against male chauvinist domestic violence have been published in Spain in an attempt to normalize, regulate, and help change mentalities.

At the turn of the millennium, in Portugal, basic and secondary education curricula were organized, with a compulsory approach to promoting sexual health and human sexuality from an interdisciplinary perspective or integrating it into subjects whose syllabus addresses this topic. Overall, the following topics were addressed: human sexuality, the reproductive system, the physiology of reproduction, AIDS and other STIs, birth control methods, family planning, interpersonal relationships, sharing responsibilities, and gender equality (Decree-Law no. 259/2000) [73]. Thus, schools adopted a cross-cutting approach to sexuality education, involving students, parents, and guardians and their associations, and investing in faculty training [51]. After 2002, emergency contraception was also provided in the consultations (Law No. 12/2001) [80], with the rapid evolution of the legislation on sexuality education as a component of health promotion and consolidation of sexuality education in the school community [46]. In particular, Order No. 25 995/2005 [74] determined that the schools must include the area of Health Education in their educational project, the close coordination between schools and primary health care, and the creation of a reference framework for the implementation of student care and support offices in secondary schools. However, it was Order 15987/2006 [75] that established new guidelines to clarify the place of sexuality education in Portuguese schools, which became one of the four components of the Health Education Project that all schools must elaborate and implement and was coordinated by a teacher appointed by each school for this purpose.

The most clarifying Portuguese legislative instrument on sexuality education was created in 2009 [77]. This law established the objectives of sexuality education in schools and its curricular and organizational framework, and the importance of promoting the school–family relationship. Reaffirming the compulsory nature of sexuality education, this law foresaw that every child and youth should have a minimum number of hours of sexuality education in each grade. A key aspect of the implementation of sexuality education in schools is the creation of health offices. According to this law, the purposes of sexuality education include addressing affectivity in a pluralistic way, focusing on equality, an informed and responsible sexuality, protection against sexual abuse, especially for the disadvantaged, and finally, the elimination of sexual discrimination. An innovative aspect of this law is that it established the figure of the teacher-coordinator, who should be the professional responsible for the introduction, supervision, and implementation of sexuality education in schools.

In 2012, the non-disciplinary curricular areas were extinguished, which was a significant barrier to the implementation of sexuality education and other components of the health education program. Sexuality education was integrated into several curricular subjects; however, many schools continued to have their health education projects, either in curricular or extracurricular activities.

In 2015, user charges were eliminated up to the age of 18 years so that their cost would not prevent adolescents from using the services [81].

In Spain, in 2013, after another political change, sexuality education disappeared from the educational agenda. LOMCE [62] did not cover aspects such as equal opportunities between men and women or sexuality; therefore, at the national level, sexuality education was not being offered in schools but was being implemented in some autonomous communities. Sexuality education was only taught in the formal context by certain teachers during the hours assigned to tutoring, with the responsibility being delegated to each center and depending on the teacher’s interest and will as to the approach and content addressed. Overall, these contents had a heterosexual approach that limits sexuality to the reproductive stage and directs the information exclusively towards risk prevention, especially unintended pregnancies, STIs, and HIV.

On the other hand, Organic Law 3/2007 [59], on gender equality between women and men, mentioned the different dimensions of SRH. Later, one of the objectives of the 2008–2011 Equal Opportunities Strategic Plan [60] was to improve women’s health and impact gender-sensitive illnesses. It proposed the inclusion of the prevention of heterosexual transmission of HIV/AIDS and sexual and reproductive health programs in the National Health System’s services, implementing information and sexuality education interventions and ensuring the prevention of unintended pregnancies and the access to adequate and effective birth control methods.

Moving forward in time, Organic Law 2/2010 [61], on Sexual and Reproductive Health and the Voluntary Interruption of Pregnancy, referred to affective-sexual and reproductive education within the formal education system. The measures to be implemented in the educational are were described in Chapter III, Article 9, of this Law: “the education system shall include training in sexual and reproductive health, as part of the comprehensive development of the personality and the training in values, including a comprehensive approach that contributes to: (a) The promotion of a vision of sexuality based on equality and co-responsibility between men and women, with special attention to the prevention of gender violence, aggression, and sexual abuse; (b) The recognition and acceptance of sexual diversity; (c) The harmonious development of sexuality based on young people’s characteristics; (d) The prevention of sexually transmitted diseases and infections, particularly the prevention of HIV; (e) The prevention of unintended pregnancies, within the scope of a responsible sexuality; (f) In the inclusion of sexual and reproductive health and health education in the education system, the reality and needs of the most vulnerable social groups or sectors, such as persons with disabilities, shall be considered, providing, in all cases, accessible and age-appropriate information and materials to these students”.

The introduction of sexual health services in the portfolio of public systems varies greatly from one region to another. In most cases, the health system’s approach to sexual health is limited to preventing STIs/HIV and unintended pregnancies, leaving aside the perspective of sexual and gender rights and sexuality care. Thus, for example, the promotion of sexuality is not integrated into all basic health. This is also closely associated with a limited strategic capacity to establish sexuality education policies in the educational domain [82]. There were some initiatives in the autonomous communities to address CSE [83], some in the educational domain, and others from non-educational bodies and organizations. One of the initiatives in the educational domain was the 2017–2021 Coeducation Plan for the schools and communities of Navarre: the SKOLAE Program. The main objective of this Coeducation Plan, which will be progressively implemented throughout the education system of Navarre, is that all students acquire basic skills to “Build their life project, based on freedom and diversity of options, without gender conditions, learning to identify inequalities, to fight against them and to exercise their right to equality within the scope of their culture, religion, social class, functional situation, etc.” [63] (p. 3).

An example of a proposal from a non-educational institution are the activities being developed in several primary schools in the region as “pilot” experiences, as a result of Law 4/2018 [64], for a Society Free of Gender Violence in Castile-La Mancha. After the publication of this law, several initiatives have been developed in Castile-La Mancha to promote equality in education, especially measures aimed at raising awareness of gender equality, preventing gender-based violence, and training the educational community through various activities and programs, namely the pilot project in the 2017–2018 academic year of the subject Education in Equality, Prevention of Gender-Based Violence and Diversity in Primary and Secondary Education. Training and awareness-raising interventions on affective-sexual education, gender roles, and gender stereotypes have also been developed. All these interventions have been supported by the Women’s Institute of Castile-La Mancha. Mention should be made to the II Strategic Plan for Equal Opportunities between Men and Women in Castile-La Mancha (2019–2024), whose priority areas for action include affective-sexual education and the prevention of gender-based violence [65].

In January 2021, Organic Law 3/2020, of 29 December (LOMLOE) [66], amending Organic Law 2/2006 on Education, of 3 May [58], entered into force. A large group of professionals in the education and health systems has high expectations about this law. Its approach to gender equality through coeducation, including, among its principles and aims, the adaption of affective-sexual education to students’ level of maturity and the prevention of gender violence, is innovative and establishes the basis for training at all stages of compulsory education.

In Portugal, for full clarification of the subjects to be included in the teaching and promotion of sexuality education, Regulation No. 196-A/2010 [78] established the contents of sexuality education for each cycle of basic and secondary education. In addition, progress is being made regarding the consolidation of the work developed with children, especially 1st-cycle children, on themes related to gender and sexuality. Article 13 of this law established the evaluation of the implementation of sexuality education in schools, which has globally been rated as good or very good [51]. In 2012, the non-disciplinary curricular areas—Civic Education, Project Area, and Accompanied Study—were eliminated, constituting a significant obstacle to implementing both sexuality education and the other components of the health education program.

However, health education projects, health education offices, and health education teacher–coordinators continued to exist in schools. These activities are often integrated into classes and extracurricular activities in cooperation with health centers, particularly the Community Care Units in the primary health care system. In Portugal, the Community Care Units were created in 2009 as functional units integrated into primary health care. They are composed of multidisciplinary teams (nurses, psychologists, nutritionists, physiotherapists, oral hygienists, and social workers) that establish protocols with the schools to implement health promotion and health literacy projects throughout the school year. These projects are developed within the scope of the National School Health Program of the General Directorate of Health, which includes areas of intervention such as health education and SRH, sexual orientation, and gender identity, among other topics. It took, in fact, 25 years for the implementation of sexuality education to go from being a mere legislative intention foreseen in the Law of 1984 to actually being implemented, with Law 60/2009 [52].

In Portugal, as an initiative to support sexuality education and fight against sexual and gender-based violence, the Family Planning Association was founded in 1967 to help people make free and informed decisions concerning their sexual life and reproductive health and promote positive parenting. It is a private institution of social solidarity (IPSS, in Portuguese) for health purposes, recognized as a non-governmental development organization (NGDO) and as a family association. The Family Planning Association is the Portuguese member of the International Planned Parenthood Federation (IPPF), a federation that brings together family planning associations from almost every country in the world. Its commitment extends to establishing national and international cooperation partnerships and protocols, focusing on regular participation in community research, education, and advocacy projects.

There is a need to invest in SRH promotion among higher education students, considering the gender perspective [84].

Since 1984, the Portuguese government has been assuming its obligations and promoting concrete measures to operationalize these rights, reinforcing the protection of maternity and paternity, introducing training and information on human sexuality in school curricula, and creating family planning consultations in health care services to provide free contraceptives and develop protection measures against STIs.

### 4.2. Education Models and Incorporation of the Gender Perspective

Until reaching the CSE model, programs, projects, and interventions about sexuality education in the formal educational domain have gone through different models and approaches throughout their history: from the moralizing model (focused on controlling sexual practice through abstinence, understood both as a preventive strategy and as an ideal for young people’s sexual conduct) to the so-called biologicist model (focused on the transmission of knowledge and limiting sexuality education to reproduction and sexual intercourse, to the medical aspects of prevention) until reaching the current model, that is, the CSE model [85]. This model incorporates the gender approach, which, according to Morgade [86], is the one that “has more relevance to the work on sexuality issues at school.” Sexuality education with a gender focus during childhood and adolescence helps young people preserve their reproductive health, promotes respect for diversity, reduces gender violence and gender inequality, and strengthens their self-esteem and the proper management of pleasure [87].

In the same line, Kirby et al. [88] and Montgomery and Knerr [89] advocate that programs can solve problems such as STIs, HIV/AIDS, and unintended pregnancies and should focus on improving affective-sexual interpersonal relationships. On the other hand, it is important to prevent sexual violence by addressing sexual risk behaviors [90] and promoting gender equality to prevent potential problems in interpersonal relationships among young people.

In this regard, we believe that sexuality education should also provide importance to one’s own and others’ sexual well-being and gender-related affective and sexual aspects, thus preventing gender-based violence [91]. Many studies leaving out the approach to sexuality-related emotions and attitudes, Lameiras et al. [92] state that sexuality education in most schools focuses on preventing pregnancy and STIs. Moreover, Fernández Costa, Juárez Martíne, and Díez David [25] also concluded that HIV/AIDS prevention programs predominate over affective-sexual programs in almost all autonomous communities in Spain. Almost the same situation occurs in neighboring Portugal, with Sousa, Soares, and Vilar [48] highlighting the need to explore the impact of sexuality education on the use of contraceptives, STIs, and sexual attitudes and behaviors of young people at Portuguese schools.

However, other contents should be incorporated into affective-sexual education: topics related to ethical values and pregnancy from a vision of knowledge rather from one of prevention, the different contraceptive methods besides the male condom (e.g., the female condom), and the myths and false beliefs about sexuality, among others [92].

The characteristics of the programs mentioned above are discussed below.

The following proposals lack content on gender equality. The Sex Education Program in the Acquisition of Knowledge about Contraceptive Methods and Emergency Contraception [33] only focuses on contraceptive methods, their use, emergency contraception, myths, and STIs. The model proposed by Rocha and Duarte [53] examined the factors facilitating the implementation of sexuality education in Portuguese schools. The inclusion of the perceptions of students, parents, and teachers is highlighted as necessary. The ¡Cuídate! program [38,39,40] is an intervention to reduce sexual risks (unintended pregnancies and STIs). Matos et al. [51] analyzed the degree of compliance with the legislation on sexuality education programs. In these programs, sexuality education is presented in the same way year after year, focusing only on the biological aspects of reproduction and STIs, which ends up making students less involved in the development of the programs. They expressed their desire to participate in this process as guides in informative and formative activities with younger peers. The COMPAS program [31,35,38,39,40] consists of four modules: information and cognitive restructuring, social skills training, problem-solving training, and maintenance strategies addressing self-instruction and covert behavioral rehearsal. The SOMOS program [93] and the P.E. Sex program [30] incorporate communication skills in their sexuality education content but do not introduce the issue of gender equality. Further, in Portugal, Almeida [47] conducted research on adolescents to characterize them in terms of sexual context variables, identify their level of knowledge, attitudes, and their motivation towards sexuality. This study also does not include a gender perspective. In the SOMOS program [93], sexuality is addressed considering biological, psychological, socio-cultural, and ethical dimensions, as well as affective issues ranging from the most basic emotions, such as desire and attraction, to falling in love. The P.E. Sex program [30] further explores the communication skills concerning sexual issues and promotes respect and responsibility in interpersonal relationships. In this context, Ribeiro, Pontes, and Santos [49] analyzed the impact of a classroom-based sexuality education program on adolescents’ attitudes towards sexuality. These authors addressed adolescents’ association between sex and pleasure, permissiveness towards casual sex without commitment, instrumentality or physical pleasure, and permissiveness with love. These themes were addressed without including a gender perspective.

The content of two sexuality education programs addressed gender equality. The “Face to Face, Heart to Heart” [36] is an STI/HIV prevention program that works separately with girls and boys and has a gender perspective. It develops knowledge and skills to promote gender equality, increase positive attitudes towards condom use, reduce fear of negotiating condom use, and ban sexist stereotypes and power in intimate relationships. In short, it promotes gender equality so that adolescents learn the difference between healthy and unhealthy relationships (those in which there is an unequal distribution of power) and recognize the difference between abuse and respect. In Portugal, Gama and Anastácio [6], in a study with a gender perspective, aimed to contextualize the meanings of sexuality and gender. It extends the concept of sexuality to include not only biological but also social and affective aspects. The authors found it important to include gender in sexuality education programs and discussions about gender-based violence, gender equity, gender stereotypes, and sexual violence, among others. In the same line, the Neither Ogres nor Princesses [32] program addresses affective-sexual education based on health and pleasure, promotion of self-esteem and autonomy, freedom of choice, equality between men and women, and respect for sexual diversity. Escalhão [50] aims to implement informed and safe decision-making along with the adoption of healthy behaviors. In this study, the author seeks to clarify and demystify concepts and situations related to contraception, STIs, affectivity, and gender roles to enable students to identify attitudes and/or behaviors related to stereotypes, sexism, and gender inequalities. Authors such as Silva and Carvalho [52] refer to a model in Portuguese schools in which sexuality education is a fundamental right to be respected and implemented from a holistic and inclusive approach to the body and the emotions. According to this model, children and young people are able to construct and give meaning to their bodies, something forgotten at school. To achieve this, the complexity of the four components of sexual identity—biological sex, gender identity, gender roles, and sexual orientation—must be integrated and addressed.

### 4.3. Study limitations

A limitation of this study was the scarcity of publications on educational programs in sexuality education applied in both countries, perhaps due to the lack of publications on this topic or limitations of the research itself. Another limitation of that the sample consisted only of adolescents attending schools.

## 5. Conclusions

Spain and Portugal are countries with a similar history but with geographical and demographic differences. Sexuality education, family planning, and reproductive health have received special attention from society in recent years as part of a progressive affirmation of citizens’ rights to education and health. In Portugal, there has been a continuous evolution of sexuality education and incorporation of the gender perspective in the legislative frameworks. In Spain, the legislation has suffered advances and setbacks depending on the governments’ political ideology. Some measures that were already outlined in previous laws have now been implemented in the legislation in force. Portugal and Spain have a law on sexuality education in schools, with compulsory and cross-cutting implementation.

However, in both countries, sexuality education among adolescents is limited, in the best circumstances, to isolated lectures or workshops given by professionals from outside the school, such as psychologists, health professionals, higher education professors, etc. Despite all the educational legislation supporting it, affective-sexual education is still part of the schools‘ unintended lessons. Furthermore, the approach to it is far from comprehensive, having mostly a biological and preventive focus.

The approach to sexuality education with a gender perspective in the educational systems in Spain and Portugal is present but not fully incorporated, with different manifestations in both countries. It is an ongoing, non-consolidated process that is sensitive to the flow of political-legislative cycles. This situation perpetuates and legitimizes the still prevailing hegemonic social model (also educational and health): the patriarchal system.

In this model, the biological dimension is the focus of the approach to sexual health, leaving the psychological and social dimensions in the background. Considering that gender is a social construct, it is necessary to work on this level so that education is comprehensive and incorporates the tools that facilitate the identification of gender inequalities (power relations) and their management.

## Figures and Tables

**Figure 1 ijerph-18-11921-f001:**
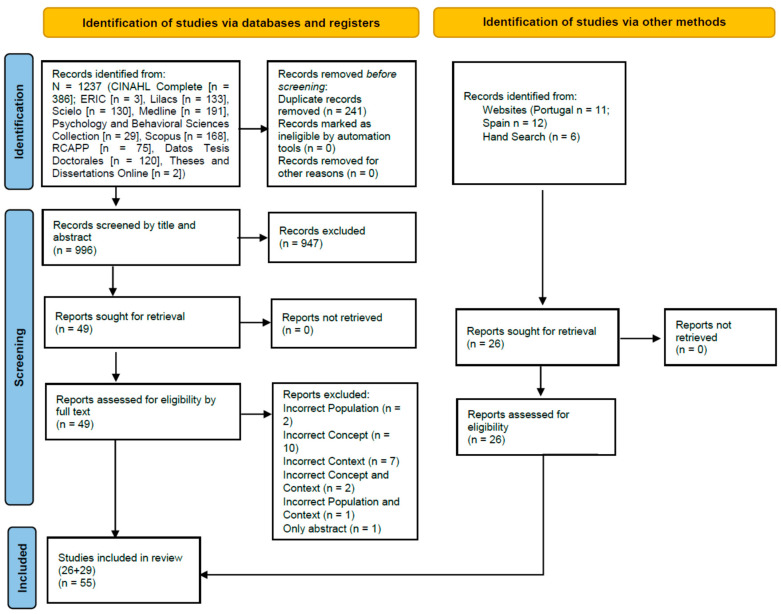
PRISMA 2020 flow diagram for new systematic reviews, which included searches of databases, registers, and other sources [23].

**Table 3 ijerph-18-11921-t003:** Main legislation in Spain.

Legislation	Content	Comments
Organic Law 8/1985 (LODE) [55]	It regulates the right to education.	No reference to sexuality education.
Organic Law 1/1990 of 3 October (LOGSE) [56]	General organization of the education system.	Sexuality education in the different educational stages and areas on a cross-cutting basis. No reference to gender.
Organic Law 10/2002, of 23 December (LOCE) [57]	Quality of education.	Elimination of sexuality education. An exclusively reproductive vision of sexuality. This law was never implemented.
Organic Law 2/2006, of 3 May (LOE) [58]	Education.	Gender equality approach through coeducation, prevention of gender violence, and respect for affective-sexual diversity. Inclusive and non-sexist perspective in secondary education. The subject “Education for Citizenship” is incorporated. It maintains sexuality education as a cross-cutting subject (as in LOGSE).
Organic Law 3/2007 of 22 March [59]	Equality between women and men.	It refers to the different dimensions of sexual and reproductive health: motherhood, fatherhood, leaves, sexual harassment.
Strategic Plan on Equal Opportunities 2008–2011 [60]	Two of its objectives are to improve women’s health and address gender-sensitive diseases.	Inclusion of prevention of heterosexual transmission of HIV/AIDS and sexual and reproductive health programs in the services provided by the National Health System.
Organic Law 2/2010 of 3 March [61]	Sexual and reproductive health and the voluntary interruption of pregnancy.	Sexuality education is included as a subject recommended by the Spanish State. It refers to affective-sexual and reproductive education in the formal context of the education system.
Organic Law 8/2013 of 9 December (LOMCE) [62]	It improves the quality of education.	It does not cover issues such as equal opportunities for men and women or sexuality. For this reason, sexuality education was not addressed in classes.
Coeducation Plan 2017–2021 for schools and educational communities in Navarra [63]	SKOLAE Program	It incorporates “learning about sexuality and proper treatment (...), self-knowledge towards the construction of relationships and love based on acceptance and respect for diversity, free from chauvinist violence.”
Law 4/2018 of 8 October [64]	For a Society Free of Gender Violence in Castile-La Mancha.	It introduces affective-sexual education on a cross-cutting basis in the syllabus of all subjects, elements that value equality, promoting women’s visibility, and, in history, the origin, development, and achievements of the feminist movement and women’s history.
II Strategic Plan for Equal Opportunities for Men and Women in Castile-La Mancha (2019–2024) [65]	Equal Opportunities Strategy.	It is made explicit that two of the priority areas for action are affective-sexual education and gender-based violence prevention.
Organic Law 3/2020 of 29 December (LOMLOE) [66]	Amending Organic Law 2/2006 on Education (LOE), of 3 May.	It adopts a gender equality approach through coeducation, including, among the principles and aims of education, the adaptation of affective-sexual education to students’ level of maturity and the prevention of gender violence. It also includes affective-sexual education in the compulsory subject of Health Education and as a cross-cutting subject, both in primary and secondary education.

**Table 4 ijerph-18-11921-t004:** Main legislation in Portugal.

Legislation	Content	Comments
Law 3/1984 of 24 March [67]	Sexuality education and family planning.	1st specific Portuguese law in the area of sexuality education in the school community.
Ordinance 52/1985, Ministry of Health [68]	It approves the regulation of family planning consultations and youth care centers.	A starting point for the interaction between health and education services concerning sexuality education.
Law no. 46/86 of 14 October [69]	Basic law of the educational system.	Sexuality education represents a new cross-cutting area in Personal and Social Development.
Decree-Law no. 286/89 of 29 August [70]	It creates the subject of Personal and Social Development in the curricula of School-Area, non-disciplinary.	Each school is responsible for Personal and Social Development.
Order 172/ME/93 of 13 of August [71]	The support program is an instrument that can be used by the schools, particularly in their response to problems and needs.	Prevention of drug addiction and AIDS, sexuality, and development of social skills, among others.
Law 120/1999 [72]	Revision of Law 3/84, for both ministries, Health and Education.	Gender equality; the need for understanding STIs and free condom distribution.
Decree-Law no. 259/2000 of 17 October (regulation of Law 120/1999) [73]	The school as a component institution to integrate sexual health promotion strategies in the curricula and the organization of curricular enrichment activities.	It promotes the school–family relationship (article 1). It includes sexuality education in the curricula of basic and secondary education integrated into health education. Gender equality is addressed.
Order no. 25 995/2005 (2nd series) of 16 December [74]	Schools are obliged to include the area of health education in the educational project.	Creation of a geographically balanced and efficient network of educational, social, and psychological resources to support schools and teachers.
Order 15987/2006 [75]	Sexuality education in schools becomes one of the four components of the Health Education Project.	It will be coordinated by a teacher appointed by the school for Health Education (Order no. 2506/2007) [76].
Law no. 60/2009—Diário da República (Official Portuguese Journal) no. 151/2009, Series I of 2009-08-06 [77]	It establishes the regime for the implementation of sexuality education in school settings.	Contents and workload of sexuality education foreseen in the law.
Ordinance no. 196-A/2010 Diário da República (Official Portuguese Journal) no. 69/2010, 1st Supplement, Series I of 2010-04-09 [78]	Regulates Law no. 60/2009 of 6 August.	Establishes sex education in basic and secondary education establishments and defines the respective curricular guidelines suitable for the different levels of education.

## Data Availability

Excluded this statement.

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
