# Peer review of "The Integration of Gender Perspective into Young People’s Sexuality Education in Spain and Portugal: Legislation and Educational Models"

_ijerph, 2021, doi:10.3390/ijerph182211921_

Round 1

Reviewer 1 Report

The authors present a broad and careful “scopy review” with the aims to explore the approach to Sexuality Education in the education of young people in Spain and Portugal, considering the incorporation of a gender perspective, and analyze the legislation in both countries. Overall it's a great review job. Despite the above, some issues, which I detail more specifically below, could improve the quality of this manuscript for its academic publication:

INTRODUCTION

Line 68

“the biological-hygienist approach”

Please explain what this model consists of and make a bibliographic reference to it.

Line 78

“condom”

Please would it please be possible here and in the rest of the text use a less colloquial and more formal term like "preservative".

RESULTS

Line 225

“the SOMOS Sexuality Education program”

What does the acronym “SOMOS” mean?

Like the “COMPAS” program (Competencies for Adolescents with a Healthy Sexuality)

Ine 249

Tabla 1

Climent et al., 2009:

To describe the predominant approaches in the sexual education of adolescents who got pregnant in the context of a particular gender socialization and its relationship with some reproductive  behaviors”

Please explain the objective better because it is confusing.

Line 264-265

“Four studies addressed the application of Sexuality Education programs [47-50].   263 These studies assessed the effects of Sexuality Education programs, namely increased 264 knowledge about family planning [47], STIs [47,48] and contraceptive methods [48], and 265 more liberal attitudes towards sexuality [49]”.

It would also be necessary to explain what the article referenced with the number 50 consisted of, in the same way that 47-48 are explained.

Line 281

Tabla 2

Siqueira& Netto, 2018 [45] y Gama&     Anastácio, 2019 [6]

It would be convenient to expand the columns of the Table a little more to avoid mixing some concepts or words with others

Line 311

It also aims to include the prevention of heterosex ual transmission of HIV/AIDS …”

I do not understand well that prevention only refers to the heterosexual sphere, HIV / AIDS prevention is not a question of sexual orientation.

Maybe the translation from place to error

Line 418

Table 4

Ordinance no. 196-A/2010 Diário da República (Official Portuguese Journal) no.

69/2010, 1st Supplement, Se- ries I of 2010-04-09 [78]”

As with the rest of the laws in Table 4, the Comments must be specified

DISCUSSIONS

Line 420

“Legislation”

Since many of the ideas that are exposed here have also been mentioned in the section "Legislation in Spain and Portugal", they should try to summarize this section much more and focus only on what has not been previously exposed, because there are times that the reading the text becomes repetitive and can confuse the reader

Line 680

It is said that “three of the proposals lacked a gender perspective, but in reality the authors speak of five: The Sex education Program (33), Cuídate (38-40), SOMOS (93), COMPAS ([31,35,38-40) and P.E. SEx Program (30)

Please clarify this.

On the other hand, I cannot find the relationship between what was done by Hernádez-Martínez et al., (33) and the proposal by Rocha and Duarte (53)

 Line 699

The SOMOS program [93] and the P.E. Sex program [30”]

I continue without being very clear when they talk about both programs (SOMOS) and P: E.Sex Program the relationship between them and the research of Almeida (47) and Ribeiro, Pontes, and Santos [49]

CONCLUSIONS

Line 761-762

“…to isolated lectures or workshops given by professionals from outside the school, such as psychologists, health professionals, teachers”

If we mention professionals from outside the school and then talk about teachers, this is an incongruity.

Please explain this better.

Line 762-763

“Despite all the educational legislation supporting it, affective-sexual education is still part of the schools’ hidden curriculum”

Please rephrase this in another way that is more understandable

Line 769-670

“This situation truly reflects the hegemonic Social Model (also educational and health): the patriarchal system”.

Perhaps it would be more correct to write that this situation perpetuates and legitimizes the still prevailing hegemonic social model, the patriarchal system

REFERENCES

Number 9

Check the link because it fails

Number 17

Check the link because it says "page not found"

Number 67

Corresponds to the law Lei No. 46/86, please review this

Number 71

Link to Portaria No. 541/98 of August 18. Check this out for

Author Response

Dear reviwers

We send the contribuitions.

Grateful for the best atention.

Reviewer 2 Report

I’m sure that authors will meet these criteria for improvement

The introduction should clarify from which educational models sexual education is explained in both countries.  Adaptive and / or inclusive models or conservative models?

Please clarify:

This does not appear in section 2.2. for what reasons did they use those keywords and not others?

Then they must analyze the effects of this model on the legal regulation that affects education.

As the period is very wide. It will be very interesting if the results indicate the evolution of the topics. It is important to present a critical evolution. The authors make it descriptive

It is an interesting article, but it’s necessary to modify some parts of the article.

Author Response

Dear reviewer 

we send the contributions.

Grateful for the best atention.

Round 2

Reviewer 2 Report

Congratulations to the authors, they have done a good job. It is an interesting paper